# Effect of HIV-1 subtype-specific Tat protein polymorphisms on Tat-TAR interaction

**Alnara Zhamalbekova[1], Mahmoud A. A. Ibrahim[2,3,4], Peter A. Sidhom[5], Syed Hani Abidi** [1]*

1 Department of Biomedical Sciences, School of Medicine, Nazarbayev University, Astana, Kazakhstan, 2 Computational Chemistry Laboratory, Chemistry Department, Faculty of Science, Minia University, Minia, Egypt, 3 School of Health Sciences, University of KwaZulu-Natal, Westville Campus, Durban, South Africa, 4 Department of Supportive Requirements, University of Technology and Applied Sciences, Nizwa, Sultanate of Oman, 5 Department of Pharmaceutical Chemistry, Faculty of Pharmacy, Tanta University, Tanta, Egypt

* m.haniabidi@gmail.com

## Abstract

### Introduction

HIV Tat protein is responsible for HIV replication activation and interacts with the TAR RNA element for its function. Genetic polymorphisms in the Tat protein sequence can affect its interaction with the TAR element. Previous studies focusing on HIV subtypes B and C found that substitutions such as C31S, R57S, and Q63E can alter the binding conformations of the Tat protein and TAR element. However, it is not known if polymorphisms in other HIV subtypes can also affect Tat-TAR interactions. This study, therefore, aims to identify subtype-specific polymorphisms in the Tat protein and their effects on Tat-TAR interactions.

### Methods

HIV Tat protein sequences from subtypes A, A1, A3, A6, B, C, D, CRF01_AE, and CRF02_A were retrieved from the HIV Los Alamos Database. The sequences were aligned and used to generate consensus sequences, which were subsequently used to identify subtype-specific Tat protein polymorphisms. The sequences were used to generate 3D models of the Tat protein using AlphaFold2, which were then used in molecular docking and molecular dynamics simulations over 200 ns using HDOCK and AMBER20, respectively, to identify the effects of subtype-specific polymorphisms on binding affinity and interaction with the TAR element.

### Results

Our results show that subtypes A6, C, and CRF02_AG have higher affinity for TAR (with binding energies of −126.8, −123.2, and −123.3 kcal/mol, respectively). On the contrary, subtypes A3 and A1 had the weakest binding affinity to TAR, with binding

provided the original author and source are credited.

**Data availability statement:** All relevant data are within the paper and its Supporting Information files.

**Funding:** This work was funded by the Nazarbayev University Faculty Development Collaborative Research Program (FDCRGP) grant No. 040225FD4710 (PI, SHA). The funders had no role in study design, data collection and analysis, decision to publish, or preparation of the manuscript.

**Competing interests:** The authors have declared that no competing interests exist.

energies of –63.3 and –57.5 kcal/mol, respectively. The increased and decreased affinity of Tat protein towards the TAR element may be attributed to subtype-specific polymorphisms (A3: K29A and R53K, A1: R57G, A6: K53R, Q54H, and V69I, C: V69I, C31S, and 58A, CRF02_AG: N24K and N29K).

## Conclusion

The results of the study suggest that subtype-specific polymorphisms can affect Tat-TAR interactions, allowing certain subtypes to interact much more strongly with TAR. This finding may have implications for the subtype-specific disease pathogenesis mediated by the Tat protein.

## Introduction

Human immunodeficiency virus type 1 (HIV-1) exhibits a high degree of genetic diversity due to its error-prone replication process and high mutation rate [1]. The genetic diversity is also responsible for the diversification of HIV-1 into nine subtypes (A-D, F-H, J, K), 159 circulatory recombinant forms, and numerous unique recombinant forms [2]. These subtypes may also differ in their transmission potential and disease severity [3 4]. HIV-B and HIV-C are the most prevalent among all HIV-1 subtypes and also the most studied [5].

HIV-Associated Neurological Disorder (HAND) is among the major diseases that HIV-1 causes [6]. The symptoms of HAND, such as memory loss, neuronal dysfunction, and brain atrophy, can be modulated by the HIV-encoded trans-activator of transcription (Tat) protein. Additionally, several other viral proteins, including Viral Protein R (Vpr), Negative Factor (Nef), Matrix Protein (MA), and gp120/gp41, also play a role in neurological disease. Furthermore, various host factors can influence HIV-1 pathogenesis; for instance, polymorphisms in the host CCR5 co-receptor can reduce resistance to HIV infection or delay disease progression [7]. Among them, Tat protein has been shown to be a major protein involved in the pathogenesis of HAND [8–10].

Tat protein is a 101 residue-long protein that consists of exons 1 and 2, which encode amino acids 1−72, and 73−101, respectively; the former contains the functional regions of the protein, and the latter contributes to viral infectivity; however, it is less crucial for HIV-1 transactivation [8 11]. HIV Tat protein recognizes and binds to a small loop structure called transactivation response element (TAR) RNA in the host by forming a Tat-TAR complex. The formation of the Tat-TAR complex is crucial for optimal HIV-1 replication [12]. Strong binding between the Tat protein and TAR increases mRNA and viral protein production; thus, the virus disseminates faster in infected hosts and results in more severe disease progression. Meanwhile, weak binding results in a restricted production of mRNA and hindered viral replication, which leads to less severe disease symptoms [12]. When the complex with strong binding affinity is formed, the virus starts to affect the normal cells, and those HIV-infected cells begin to actively produce neurotoxic viral particles that can attack the

immune cells [13]. Tat protein can induce virus-infected macrophages and microglia to release toxic substances by acting like a neurotoxin [8].

Like other HIV-1 proteins, Tat exhibits genetic polymorphisms that can significantly affect viral pathogenesis and disease progression by altering the conformation of the Tat-TAR complex and/or the binding affinity between these molecules [14]. Recent research has highlighted the significance of natural polymorphisms of the Tat protein sequence, particularly mutations at key residues such as C31S, R57S, and Q63E, in influencing the development of HAND [5]. Studies on subtypes B and C have shown that these polymorphisms can alter the binding affinity of the Tat protein to the TAR element, with subtype-specific variations affecting the neuropathophysiology of HAND [6]. According to previous studies, the Arginine residue at position 57 is crucial for effective transactivation and can lead to a higher degree of neuropathogenesis. Subtype C has a mutation at this site, and substitution by Glycine reduces the transactivation; thus, it is less pathogenic than subtype B [15,16]. Furthermore, previous studies have shown that the basic domain in the Tat protein sequence, amino acids 47−58, is essential for TAR binding, and mutations at this site can influence Tat-TAR interaction [17]. Substitution of Lysine, for example, has been shown to significantly reduce the binding of the Tat protein to TAR, whereas substitution of Arginine lowers the binding affinity to a lesser extent [14]. Likewise, it was found that wild-type amino acids K (Lysine), R (Arginine), and Q (Glutamine) in the basic domain (48−58 positions) are crucial for strong binding and effective activation of HIV-1 genome transcription [14].

Additionally, several studies have demonstrated that the Tat protein is also important in the establishment and reversal of viral latency. For example, a study done by Geng et.al (2016) found that Tat mutations such as V36A, Q66A, V67A, S66A, and S77A have such abilities as reversing the latency of the virus, while another study by Kamori & Ueno (2017) concluded that polymorphisms such as P10S, W11R, K19R, A42V, and Y47H, C22S also can contribute to latency by decreasing the transaction activity [18,19]. Although the role of Tat-TAR binding in HIV transcription, latency, and reactivation has been extensively studied, however, how genetic variations among Tat sequences of different subtypes (other than subtypes B and C) affect their interactions with TAR has not been investigated. This leaves a gap in understanding how Tat protein polymorphisms associated with different non-B, non-C HIV-1 subtypes can affect Tat-TAR interactions, potentially contributing to HAND development across different viral strains [14]. In our paper, rather than challenging the necessity of the Tat-TAR axis, we investigate the effect of polymorphisms on their interaction and how they might contribute to transactivation efficiency and clinical disease progression.

To address these gaps, the current study employed sequence analysis, 3D structure prediction, molecular docking, and molecular dynamics simulations to identify subtype-specific Tat protein polymorphisms, followed by analysis of their effects on the Tat-TAR interaction.

## Materials and methods

### Retrieval of HIV subtype-specific sequences, sequence alignment, synthesis of consensus sequence

All available HIV-1 Tat amino acid sequences (110 amino acids long; S1 Table) were retrieved for subtypes A (n = 3), A1 (n = 1204), A3 (n = 3), A6 (n = 236), B (n = 21000), C (n = 12124), D (n = 397), CRF01_AE (n = 3568), and CRF02_AG (n = 330) from the Los Alamos HIV Sequence Database (https://www.hiv.lanl.gov/content/index) [20]. Out of all HIV subtypes, these nine subtypes were selected because of their high global prevalence worldwide (https://www.hiv.lanl.gov/components/sequence/HIV/geo/geo.html). The downloaded HIV sequences were edited and used to generate subtype-specific HIV-1 consensus sequences with the "Simple Consensus Maker" tool (https://www.hiv.lanl.gov/content/sequence/CONSENSUS/SimpCon.html). Consensus sequences were built using a threshold of >50% amino acid frequency at each position, representing the most prevalent residues within each subtype. This approach allows a useful model for comparative subtype analysis, capturing functionally significant variants rather than intra-subtype variations. A consensus-based approach demonstrated substantial sequence conservation across all 9 subtypes: 90% of amino acid residues met the 50% threshold, ranging from 60–95%. The remaining amino acid residues (10%) exhibited variability, but

none of these polymorphic sites corresponded to previously investigated polymorphisms that influence Tat-TAR binding affinity.

### Secondary structure prediction

The secondary structure for each subtype-specific Tat protein was predicted using the PSIPRED (http://bioinf.cs.ucl.ac.uk/psipred/) tool [21]. This tool was used to identify subtype-specific changes in secondary structure and also to identify the physicochemical properties of amino acids.

### Retrieval of TAR structure, generation of subtype-specific 3D models of Tat proteins, quality assessment, and refinement of tertiary structures

The 3D structure of TAR RNA (PDB ID: 6xh2) was downloaded from the protein data bank. This structure was selected based on recency (the year 2020), highest resolution (X-ray diffraction: 1.71 Å), and similarity with all other structures resolved using X-ray diffraction, and most (9/14) structures resolved using NMR. In the analyses, a single, highly conserved TAR was used across all HIV-1 subtypes to ensure that observed differences in Tat-TAR binding were due to Tat polymorphisms rather than TAR sequence variation. The 3D models of subtype-specific Tat proteins were generated using AlphaFold2 software [22–24]. For this, the consensus Tat protein sequences for each subtype were used as input, and five models were generated and ranked by plDDT (predicted local distance difference test) score [25]. The model with the highest confidence was selected. The 3D Tat protein structures were further refined using the GalaxyRefine2 tool in the GalaxyRefine web server (https://galaxy.seoklab.org/cgi-bin/submit.cgi?type=REFINE2) [26]. The refined models were checked for quality using the Verify 3D tool in SAVES (https://saves.mbi.ucla.edu/). As a result of this assessment, all investigated subtypes scored 80% and higher, suggesting they were reliable for further steps.

### Molecular docking, molecular dynamics simulations, and Tat-TAR interaction analysis

The molecular docking approach was used to analyze Tat-TAR binding and interactions using the HDOCK server (http://hdock.phys.hust.edu.cn/). HDOCK enables the docking of Tat protein and TAR element based on hybrid algorithms and provides various conformations of possible bindings by ranking them based on the implemented docking scores [27]. For docking, refined subtype-specific Tat protein structures were uploaded as receptor molecules in PDB format, while the TAR RNA structure was uploaded as a ligand in PDB format. The binding residues were also specified, including receptor binding site residues 48−58 in the chain A of the Tat protein and ligand binding sites residues +23-+25 in the TAR element [6]. In the HDOCK, after docking, the complexes exhibiting the lowest docking scores and high confidence scores of > 0.7 (< 0.5 = complex formation is unlikely; 0.5–0.7 = probability of complex formation is high, > 0.7 very high likelihood of complex formation [27]) were selected for further analysis using molecular dynamics simulations (MDS). Tat-TAR interactions were then analyzed using the Discovery Studio Visualizer v21.1.0.20298 [28]. MDS can be considered more reliable than molecular docking, as a) MDS allows both protein and ligand to be dynamic, providing a more realistic representation of molecular interactions; b) MDS also represents the dynamic behavior of the complex over time, including conformational changes and stability [29,30].

MDS was performed to investigate the dynamic behavior of the Tat-TAR complex, simulating its behavior in a realistic environment. In this study, MDS was performed on Tat protein poses from different subtypes exhibiting the highest affinity for the TAR element using AMBER20 [31]. Detailed MDS methods are available elsewhere [32–34]. Briefly, the AMBER force field 14SB and the RNA.OL3 was utilized to parameterize the Tat protein and TAR RNA, respectively [35,36]. All systems analyzed were immersed in an octahedral box containing TIP3P water molecules with 1.2 nm marginal radii [37]. Na$^+$/Cl$^-$ ions were added to neutralize the solvated complex. Minimization was conducted to eliminate steric clashes, using the steepest descent and conjugate gradient algorithms for 5,000 iterations. During the heating step, the temperature

was gradually raised to 310 K over 50 ps using the Langevin thermostat. Subsequently, the equilibrated systems underwent a 10 ns equilibration phase under the NPT ensemble. The SHAKE algorithm, with a 2 fs integration step, was used to constrain all bonds involving hydrogen atoms. The temperature was held at 310 K using a Langevin thermostat, while pressure was controlled with a Berendsen barostat [38]. The production phases were executed over 200 ns (all subtypes) and 500 ns (strongest and weakest binders only) MDS, with atomic coordinates saved every 10 ps, resulting in 20,000 and 50,000 trajectories, respectively. This MDS duration (twice that used in comparable studies) allowed us to establish stable binding energies and obtain consistent RMDS results, confirming that the core Tat-TAR interactions were established within this timeframe. MDS were performed using the GPU-accelerated MD engine in AMBER20 (pmemd.cuda). The molecular mechanics/generalized Born surface area (MM/GBSA) approach was employed to calculate the binding energy ($\Delta G_{binding}$) [39]. $\Delta G_{binding}$ was determined by calculating the difference between the energy of the complex ($G_{Tat\text{-}TAR}$) and the sum of the energies of the protein ($G_{Tat}$) and the RNA ($G_{TAR}$), as shown in the equation below:

$$\Delta G_{binding} = G_{Tat–TAR} - G_{TAR} - G_{Tat}$$

Additionally, we calculated the median (IQR [25th, 75th]) binding energy ($\Delta G_{binding}$) to determine which Tat protein from different subtypes exhibits the highest and lowest binding affinity to TAR. The calculations were performed in JASP 0.18.30.0.

## Results

### Assessment of Tat protein sequences from nine HIV-1 subtypes

Analysis of Tat protein sequences showed subtype-specific polymorphisms at different amino acid positions, including positions 7, 12, 19, 21, 24, 29, 35, 39–40, 54, 57–58, 60–64, 67–68, 70, 74–81, 84–85, 90, 92–95 and 99–100 (Fig 1A, black font). Subtype B showed the highest polymorphisms, while the most conserved subtype, with the lowest polymorphism, was subtype A1 (Fig 1A).

Among these, sites 19, 21, 24, 54, 57, 60, 63, 70, 75, 77, 78, 81, 85, 95, 99, and 100 exhibited non-conservative amino acid changes (Fig 1B).

Analysis of the secondary structure showed that all subtypes, except C and D, exhibited helix structures at the polymorphic position 39 (Fig 2). Subtypes CRF01_AE, A3, and B also showed helix structure at polymorphic position 35 (Fig 2). Helices were observed at position 90 for all subtypes and at position 92, while C and D did not exhibit helices at 92 (Fig 2). Helical structures were seen for subtypes CRF01_AE, A, A1, A6, and B at polymorphic position 93, and for subtypes CRF01_AE and A6 at position 94, and subtype A6 showed helix structure at position 95, with subtype A6 also showing a helix at position 85 (Fig 2). All other polymorphic sites exhibited coil structure.

### Molecular docking, molecular dynamics simulations (MDS), and Tat-TAR interaction analysis

The molecular docking analysis between the subtype-specific Tat protein and the TAR element showed that all Tat protein models formed stable complexes with TAR (Fig 3 and S1 Fig).

Previous studies have demonstrated that the solvent effect, conformational flexibility of ligand-protein complexes, and dynamic behavior are essential to enhance the accuracy of predicted ligand-protein binding energies [41,42]. Consequently, MDS, accompanied by MM/GBSA binding energy calculations, was conducted to investigate the interaction dynamics of Tat protein from different subtypes with the TAR element. Table 1 lists the estimated binding energies for Tat-TAR complexes over 200 ns MDS.

The MDS analysis of subtype-specific Tat protein-TAR element interactions showed that all Tat protein models formed stable complexes with TAR, with binding energies ranging from −57.5 kcal/mol to −126.8 kcal/mol. The median (IQR [25th, 75th]) binding energy was −82.5 kcal/mol (IQR: −123.3, −68.8]). The results showed that Tat proteins from subtypes A6,

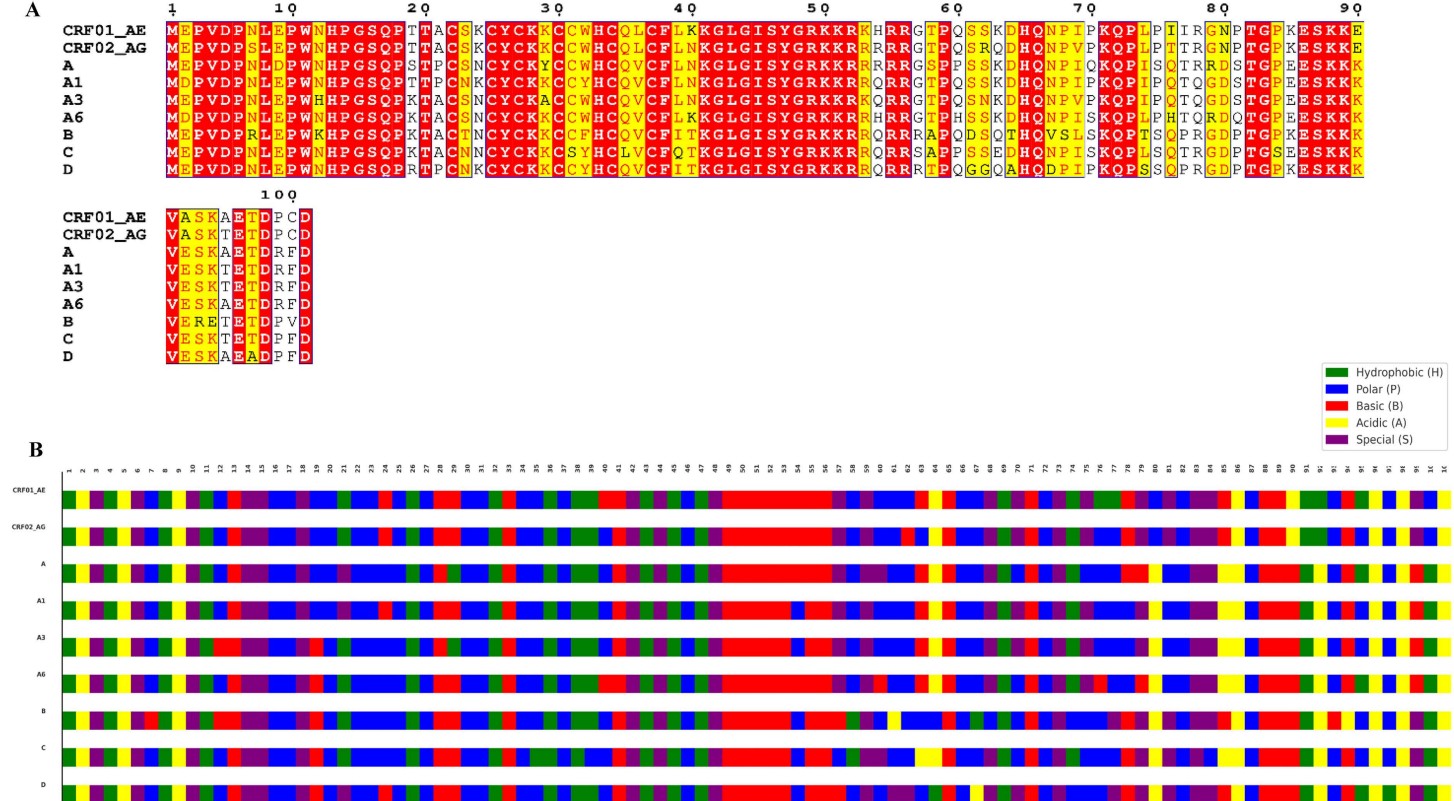

**Fig 1. Tat protein sequence alignment for the nine HIV-1 subtypes. A)** The Tat protein sequence alignment, with conserved, semi-conservative, and non-conservative sites highlighted in red, yellow, and white, respectively, while the polymorphisms are shown in black font. The alignment figure was prepared using the ESPript3 tool (https://espript.ibcp.fr/ESPript/ESPript/) [40]. **B)** The nature of amino acids in each subtype: green = hydrophobic, blue = polar, red = basic, yellow = acidic, purple = special (Glycine and proline, due to unique physicochemical properties that influence protein structure and function). The figure was generated in MS Excel.

CRF02_AG, and C had the highest binding energies ($\Delta G_{binding}$: −126.8 to −123.2 kcal/mol) with the TAR element, while the remaining subtypes, especially A3 and A1, displayed comparatively weaker binding energies ($\Delta G_{binding}$: −63.3 to −57.5 kcal/mol). To further evaluate the reliability of these observations, extended MDS of 500 ns were conducted for the A6-TAR (strongest binder) and A1-TAR (weakest binder) complexes. The MM/GBSA calculations performed over the 500 ns trajectories yielded $\Delta G_{binding}$ values of −140.6 kcal/mol and −73.4 kcal/mol for the A6-TAR and A1-TAR complexes, respectively, confirming the stronger interaction of the A6 subtype with the TAR element (S2A Fig). We also analyzed individual energy components that contributed most to the binding energy of the Tat-TAR complexes over 200 ns MDS (Table 1). The results showed that both $E_{ele}$ (ranging from −2271.8 to −5771.9 kcal/mol) and $E_{vdW}$ (ranging from −86.3 to −125.5 kcal/mol) played significant and favorable roles in Tat-TAR interactions.

## Post-dynamics analyses

Post-dynamics analyses were conducted over 200 ns of MDS to assess the energetic and structural stability of the Tat-TAR complexes. These analyses included root-mean-square deviation (RMSD) and binding energy per frame.

| Polymorphic sites | 7 | 12 | 19 | 21 | 24 | 29 | 35 | 39 | 40 | 54 | 57 | 58 | 60 |
|---|---|---|---|---|---|---|---|---|---|---|---|---|---|
| CRF01_AE | C | C | C | C | C | C | H | H | C | C | C | C | C |
| CRF02_AG | C | C | C | C | C | C | C | H | C | C | C | C | C |
| A | C | C | C | C | C | C | C | H | C | C | C | C | C |
| A1 | C | C | C | C | C | C | C | H | C | C | C | C | C |
| A3 | C | C | C | C | C | C | H | H | C | C | C | C | C |
| A6 | C | C | C | C | C | C | C | H | C | C | C | C | C |
| B | C | C | C | C | C | C | H | H | C | C | C | C | C |
| C | C | C | C | C | C | C | C | C | C | C | C | C | C |
| D | C | C | C | C | C | C | C | C | C | C | C | C | C |

| | 61 | 63 | 64 | 67 | 68 | 70 | 74 | 75 | 76 | 77 | 78 | 79 | 80 |
|---|---|---|---|---|---|---|---|---|---|---|---|---|---|
| CRF01_AE | C | C | C | C | C | C | C | C | C | C | C | C | C |
| CRF02_AG | C | C | C | C | C | C | C | C | C | C | C | C | C |
| A | C | C | C | C | C | C | C | C | C | C | C | C | C |
| A1 | C | C | C | C | C | C | C | C | C | C | C | C | C |
| A3 | C | C | C | C | C | C | C | C | C | C | C | C | C |
| A6 | C | C | C | C | C | C | C | C | C | C | C | C | C |
| B | C | C | C | C | C | C | C | C | C | C | C | C | C |
| C | C | C | C | C | C | C | C | C | C | C | C | C | C |
| D | C | C | C | C | C | C | C | C | C | C | C | C | C |

| | 81 | 84 | 85 | 90 | 92 | 93 | 94 | 95 | 99 | 100 |
|---|---|---|---|---|---|---|---|---|---|---|
| CRF01_AE | C | C | C | H | H | H | H | C | C | C |
| CRF02_AG | C | C | C | H | H | C | C | C | C | C |
| A | C | C | C | H | H | H | C | C | C | C |
| A1 | C | C | C | H | H | H | C | C | C | C |
| A3 | C | C | C | H | H | C | C | C | C | C |
| A6 | C | C | C | H | H | H | H | H | C | C |
| B | C | C | C | H | H | H | C | C | C | C |
| C | C | C | H | H | C | C | C | C | C | C |
| D | C | C | C | H | C | C | C | C | C | C |

**Fig 2. Secondary structure assessment of the Tat protein from nine HIV-1 subtypes.** The figure shows the changes in the secondary structure of HIV-1 Tat protein sequences from nine subtypes at polymorphic sites. Key: gray = C (coil), pink = H (helix). The figure was generated in MS PowerPoint.

## Binding energy per frame

The correlation between binding energy and time was estimated to evaluate the energetic stability of the Tat-TAR complexes over the 200 ns MDS (Fig 4A). The Tat-TAR complex from all subtypes, notably A6, CRF02_AG, and C, displayed significant stability throughout 200 ns MDS, with average $\Delta G_{binding}$ values ranging from −57.5 to −126.8 kcal/mol. These results further confirmed the ability of the Tat protein, especially from subtypes A6, CRF02_AG, and C, to form stable complexes with the TAR element throughout the simulation.

## RMSD analysis

The structural stability of the Tat-TAR complexes was evaluated through changes in backbone RMSD relative to the initial structures over the 200 ns MDS (Fig 4B). Most RMSD values fluctuate between 0.1 and 0.9 nm, with the Tat-TAR complexes reaching stability after the initial 20 ns. To further assess the reliability of these observations, extended MDS of 500 ns were conducted for the A6-TAR and A1-TAR complexes. The RMSD analysis of the extended trajectories showed average RMSD values of approximately 0.6 nm and 0.8 nm for the A6-TAR and A1-TAR complexes, respectively, confirming their structural stability over longer simulation times (S2B Fig). These results indicated that Tat proteins from different

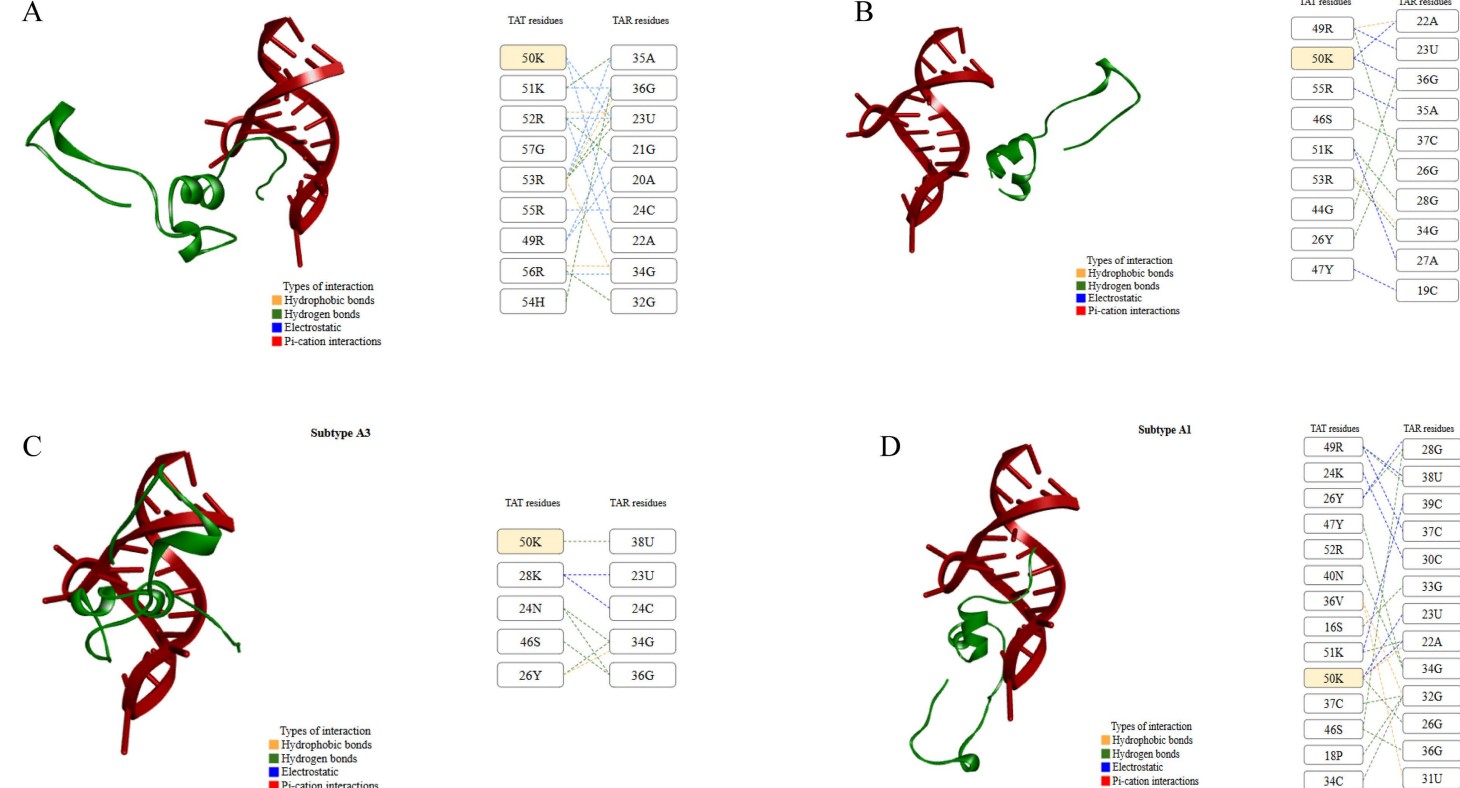

**Fig 3. The Tat-TAR interaction analysis of significant subtypes.** The figure (right) shows amino acids of the Tat protein binding to RNA nucleotides, the positions of interacting residues, and the types of interactions (hydrophobic, pi-cation, hydrogen bonds, and salt bridges). The left panel shows the Tat-TAR interaction, with the Tat protein in red and the TAR element in green. Only subtypes with the highest (A and B) and lowest (C and D) binding affinity in docking/MDS are shown. Yellow shaded boxes show common amino acids among subtypes that were involved in Tat-TAR interaction. The Tat-TAR poses were generated in Discovery Studio v21.1.0.20298 (https://discover.3ds.com/discovery-studio-visualizer-download), while interactions were drawn using MS PowerPoint.

**Table 1. Contribution of different energy components of various subtype-specific Tat and TAR complexes over 200 ns MDS and amino acids involved in Tat-TAR interactions obtained after docking using HDOCK tools. Underlined amino acids are common amino acids between subtypes.**

| HIV-1 Subtype | MM/GBSA Binding Energy (kcal/mol) | | | | | | | Interacting amino acids |
|---|---|---|---|---|---|---|---|---|
| | $\Delta E_{vdW}$ | $\Delta E_{ele}$ | $\Delta E_{GB}$ | $\Delta E_{SUR}$ | $\Delta G_{gas}$ | $\Delta G_{Solv}$ | $\Delta G_{binding}$ | |
| A6 | −110.0 | −5417.9 | 5416.1 | −15.0 | −5527.9 | 5401.1 | −126.8 | 49R, 50K, 51K, 52R, 53R, 54H, 55R, 56R, 57G |
| CRF 02_AG | −110.6 | −5416.8 | 5418.4 | −14.3 | −5527.4 | 5404.1 | −123.3 | 47Y, 26Y, 44G, 46S, 49R, 50K, 51K, 53R, 55R |
| C | −107.4 | −5771.9 | 5772.7 | −16.5 | −5879.3 | 5756.1 | −123.2 | 46S, 49R, 50K, 51K, 52R, 53R, 55R, 56R |
| D | −125.0 | −5886.4 | 5911.8 | −16.9 | −6011.4 | 5894.9 | −116.5 | 26Y, 42G, 46S, 50K, 51K, 52R, 53R, 55R, 56R, 58T |
| CRF 01_AE | −117.7 | −4943.1 | 4993.3 | −15.0 | −5060.7 | 4978.3 | −82.5 | 26Y, 41K, 46S, 49R, 50K, 52R, 55R, 56R, 59P |
| A | −116.2 | −1835.0 | 1889.8 | −15.0 | −1951.2 | 1874.8 | −76.4 | 16S, 22C, 26Y, 36V, 37C, 40N, 46S, 47Y, 49R, 50K, 51K |
| B | −86.3 | −4465.1 | 4489.1 | −12.0 | −4551.4 | 4477.2 | −74.3 | 26Y, 28K, 31C, 45I, 47Y, 50K, 52R, 55R |
| A3 | −98.2 | −2271.8 | 2319.9 | −13.1 | −2370.1 | 2306.7 | −63.3 | 24N, 26Y, 28K, 46S, 50K |
| A1 | −92.3 | −2523.2 | 2570.3 | −12.2 | −2615.5 | 2558.1 | −57.5 | 16S, 18P, 24K, 26Y, 34C, 36V, 37C, 40N, 46S, 47Y, 49R, 50K, 51K, 52R |

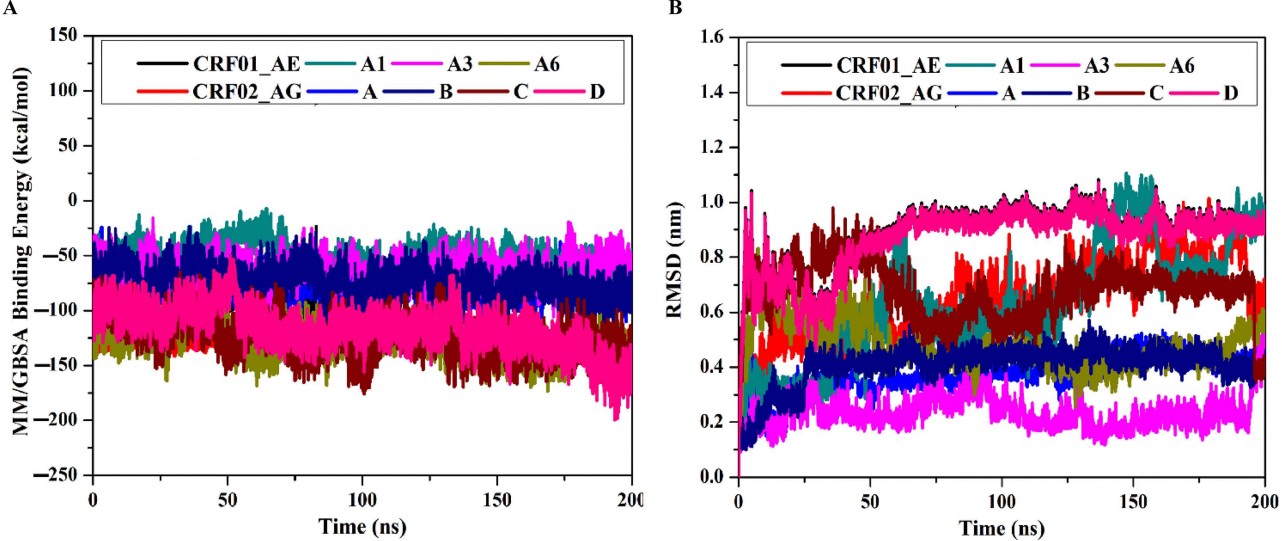

**Fig 4. Post-dynamics analyses.** The figure shows A) binding energy per frame and **B)** RMSD of the backbone atoms with respect to the initial structure of the Tat protein from different subtypes. Key: CRF01_AE (black), CRF02_AG (red), A (blue), A1 (cyan), A3 (magenta), A6 (yellow), B (navy), C (wine), and D (pink) in complex with TAR RNA over 200 ns MDS. The figures were generated using OriginPro (OriginLab, Northampton, MA, USA; https://www. originlab.com/).

subtypes, especially those from A6, CRF02_AG, and C, bind tightly to the TAR element without disrupting the overall structure.

## Discussion

This study aimed to identify HIV-1 subtype-specific Tat protein polymorphisms that may affect the interaction between the Tat protein and the TAR element. Analysis of the subtype-specific polymorphisms showed that the polymorphic sites were mainly located in the region spanning from amino acids 57−101. This is in agreement with the previous studies showing that the first exon (1−72) is more conserved than the second exon (73−101), which is more prone to mutations across the HIV-1 subtypes [43].

In the next step, we used molecular docking and molecular dynamics simulations to identify interaction sites and to determine which subtype-specific Tat protein had the highest affinity for the TAR element. Overall, our results showed that the Tat protein from all subtypes formed stable interactions with the TAR element. According to the scores of global binding energies, Tat protein from subtypes A6, CRF02_AG, and C exhibited the highest binding affinities, while Tat protein from subtypes A1 and A3 showed the least binding affinity (Table 1). Tat from A6, CRF02_AG, and C formed several interactions in the basic domain (48−58), such as 49R, 50K, 51K, 52R, 53R, 54H, 55R, 56R, and 57G (Table 1). Additionally, the RMSD values of Tat-TAR complexes for subtypes A6, CRF02_AG, and C, respectively, were in the range of 0.4–0.6 nm, 0.5–0.9 nm, and 0.5–0.9 nm, with some flexibility between 0 and 50 ns. Considering the importance of the basic domain in the interaction between Tat and TAR [44], the subtype-specific polymorphisms (subtype A6: K53R and Q54H, subtype CRF02_AG: N24K and N29K, and subtype C: C31S, 57S, and 58A) in this domain may have contributed to the high binding energy observed in the study. *Ronsard et al.* observed that Arginine in the basic domain at positions 49, 52, 53, 55, 56, and 57 is important in forming hydrogen interactions between Tat and TAR, and its mutation would lead to decreased binding in subtypes. Thus, the presence of Arginine in these subtypes within this range may have led to a stronger Tat-TAR complex formation. The high binding affinity observed in the subtype CRF02_AG may be explained

by the presence of the N24K and N29K polymorphisms, as lysine residues have been shown to play a significant role in enhancing viral Tat transactivation [45]. However, it is important to note that only a limited number of studies have explored the natural polymorphisms of this subtype, resulting in insufficient supporting data. A study by Ronsard et al. investigated the effects of individual amino acid substitutions and found that S62R/G mutations can reduce TAR binding and, consequently, impair viral transactivation. This finding does not align with the increased binding affinity of TatCRF02_AG observed in our study. Therefore, further investigation into the natural polymorphisms of this subtype is needed. On the contrary, Tat protein from subtypes A3 and A1 mainly formed interactions outside the basic domain, with very few within it, which may have contributed to decreased binding affinity [14]. Ronsard et al. observed that the presence of lysine at positions 28, 29, 50, 51, and 71 is also crucial for the effective binding of Tat protein to TAR. Substitution of Lysine with another amino acid (such as K29A and R53K in subtype A3 and R57G polymorphism in subtype A1) may have led to a decrease in binding affinity [13,14]. Our analysis showed the presence of the V69I polymorphism in both subtypes A6 and C (while subtype A3 had I69V), which may have contributed to higher Tat-TAR binding affinity, but not in subtype CRF02_AG. Since no study to date has investigated the polymorphism at position 69, this position could be the focus of future studies that specifically examine its effect on the Tat-TAR interaction. A study by Ruiz et al. (2019) found that an Arginine residue at position 57 is important for effective Tat-TAR interaction, and its mutation significantly decreased Tat uptake [46]. Our results are consistent with these findings and identified that both subtypes A3 and A1 had R57G, which could contribute to their binding affinities. However, few studies have investigated the polymorphisms of subtype A1 and the Tat-TAR interaction overall, which is a limitation. As a result, there is limited supporting data to explain how specific mutations contribute to the observed binding affinities.

Our findings align with experimental observations of Tat functional diversity. Ronsard et al. (2017) demonstrated that naturally occurring Tat variants, such as the S46F mutation in North Indian variants, exhibit distinct Tat-binding capacities and transactivation efficiencies in cell-based assays [47]. In their study, variants with S46F showed enhanced transactivation, consistent with our observations that the core basic domain significantly affects Tat-TAR binding and its energetics [48]. While their study focuses on specific mutations in specific subtypes, our analysis, based on 9 prevalent subtypes, identifies polymorphisms such as K53R, Q54H, and V69I that similarly affect Tat-mediated transcription. Also, Ronsard et al. (2019) found that TatB induced greater (53%) cell death in T cells than TatC (25–28%) [49]. This matches our computational findings, which show that subtype C binds strongly to TAR.

Additionally, the results of this analysis about Tat-TAR binding affinity variations across subtypes are consistent with broader patterns of HIV-1 genetic and functional diversification observed in other viral proteins. For instance, a study from North India reported the emergence of Vif B/C recombinants and that Vif C variants showed higher APOBEC3G-degrading activity than Vif B variants [50]. This correlates with our observation that certain Tat subtypes show enhanced Tat-TAR binding, potentially due to specific polymorphisms.

The functional significance of our findings extends beyond binding to the TAR. For example, it is known that Tat interacts with host proteins to regulate viral replication. Raja et al. (2017) have shown that Tat stabilizes the Mdm2 host protein that occurs via phosphorylation at serine-166. When serine-166 was mutated, viral replication was not supported [51]. This phosphorylation-dependent regulation could be influenced by Tat polymorphisms. Likewise, subtypes with high TAR-binding affinity could have more favourable conformations for both TAR and Mdm2 interactions. This suggests that polymorphisms that affect TAR binding may also affect broader regulatory systems involving the host protein.

It is important to mention that our results for subtypes B and C Tat protein differ from the previous studies by Williams [6], Gotora [44] and Ronsard [46], who showed that Tat from subtype B had higher binding affinity as compared to subtype C Tat protein due to high levels of flexible regions, more significant numbers of hydrogen bonds, and more negative value of free energy of binding. This difference may arise from differences in Tat protein modeling, TAR element model selection, MDS run duration, etc. For example, previous studies focused on the wild-type subtype B polymorphism, whereas we used a consensus-sequence approach, as in a study by Williams, which presented a more representative subtype B

sequence. We also used state-of-the-art AlphaFold2 software (compared to Modeller or Swiss-model in other studies) to generate subtype-specific 3D Tat protein models. AlphaFold2 software offers superior modeling as compared to other software, as it a) can predict structures without requiring template structures, b) uses advanced neural networks incorporating evolutionary, physical, and geometric constraints, c) employs multiple sequence alignments (MSAs) to detect evolutionary patterns and correlations between mutations, and d) processes both intra-chain and homotypic contacts effectively [52]. Additionally, we utilized the most recent TAR structure (PDB ID: 6xh2), with excellent resolution (X-ray diffraction 1.71 Å) and remarkable similarity to other X-ray diffraction and NMR-based structures. Finally, we used a 200 ns MDS run and confirmed a few results using a 500nm run, which offers a more detailed analysis of Tat-TAR dynamics, energy changes, and stability over time. Based on these factors, we believe our study provides a more robust and accurate analysis of the interaction between the Tat protein from different subtypes and the TAR element. Our results, however, support the studies by *Arellano* [1] and *Johri* [53], who concluded that Tat subtype C had a higher binding affinity due to 57S, 63E, 70S, and 58A [46,54], leading to more effective transactivation of the viral genome. In this case, subtype C showed greater neuropathogenicity.

The results of this study can have real-world implications. Previous studies have found that, in general, higher Tat-TAR binding affinity indicates increased transactivation and, subsequently, more severe clinical outcomes. A study by *Williams & Cloete* [6] showed that the arginine residue at position 57 (R57) in the Tat protein enhances transactivation and neuropathogenesis; thus, its substitution by glycine or serine (as seen in subtype A6, CRF02_AG, and C) typically reduces transactivation efficiency. However, the subtype C Tat protein also contains an additional substitution at position 31 (C31S), observed in our analysis, that significantly enhances Tat-TAR binding affinity and may compensate for the loss of function caused by the R57S substitution. Consequently, despite reduced transactivation due to the R57S mutation, subtype C can exhibit increased neuropathogenesis overall, driven by enhanced Tat-TAR complex formation mediated by the serine residue at position 31 [1,13,46,53]. While the high binding affinity of A6 and CRF02_AG might be attributed to the presence of amino acid residues critical for interaction, such as K28, K29, K50, K51, K71, R49, R52, and R53 [46], which formed bonds with the TAR element. To identify any correlation between disease progression and subtype-specific polymorphisms, we stratified available Tat sequences by clinical stages. We found that AIDS-stage variants from best-binder subtypes A6 and CRF02_AG showed more charged basic domains (RRHRR/RRRRR), which might enhance transactivation. While the chronic variant of CRF02_AG showed a less charged basic domain (RRQRR), suggesting that viral adaptation and its attenuation occur in long-term infection. The conservation of the Tat sequence in chronic and acute variants of subtype C suggests that disease progression might be linked to viral or host determinants. As shown in Table 2, different HIV-1 subtypes accumulate specific polymorphisms as infection progresses. For example, subtype C maintains C31S and R57S from acute through chronic stages, but new neuropathogenic variation P68L appears only in the AIDS stage.

However, in the absence of studies examining the association between Tat polymorphisms in HIV-1 non-B, non-C subtypes and neuropathogenesis, the role of these polymorphisms remains unclear and warrants further investigation.

**Table 2. Subtype-specific Tat polymorphisms in sequences from different HIV-1 clinical stages. Polymorphisms (observed in the study) are shown relative to subtype consensus sequences and were identified by aligning clinical sequences with subtype-specific consensus sequences.**

| | Polymorphisms observed in sequences from different clinical stages | | |
| --- | --- | --- | --- |
| Subtypes | Acute | Chronic | AIDS |
| CRF02_AG | N24K, N29K, 57R, K1I | N29M, R57Q, K41S | 57R, S46F, K41N |
| C | C31S, R57S, V69I | No new changes – same as seen in the acute stage | S31N, K50N |
| A6 | Not available | Not available | Q55H, R57H, T60L, T96A |
| A1 | Not available | R57Q, T96S, C38Q | Not available |

We acknowledge certain limitations inherent in our study. A primary limitation is that all findings regarding the analysis of subtype-specific Tat polymorphisms in Tat-TAR interactions are derived solely from in silico analyses and rely on predictive computational models. The observations could not be confirmed using *in vitro* or *in vivo* assays due to logistical and resource limitations. Therefore, the biological and clinical implications of these subtype-specific polymorphisms require validation through in vitro and in vivo functional assays in future studies. However, factors such as 200 ns MDS, confirmation using 500 ns MDS, selection of a recent, high-quality TAR structure, and use of AlphaFold2 for Tat protein modeling increased the robustness of our analyses.

In conclusion, the current study found that subtypes A3 and A1 exhibit the lowest binding affinity, as measured by free energy of binding. The likely reason for the decreased affinity is the K29A, R53K, and R57G polymorphisms. On the contrary, subtypes A6, CRF02_AG, and C exhibited the highest binding affinity to TAR, which could be due to K53R, C31S, N24K, N29K, and 70S. Subtype-specific polymorphisms may contribute to differences in disease (HIV-associated neurological disorders) dynamics, and thus, it is important to confirm the effect of subtype-specific polymorphisms using functional assays.

## Supporting information

**S1 Table. Accession numbers of the HIV-1 Tat sequences used in this study.**
(XLSX)

**S1 Fig. The Tat-TAR interaction analysis of significant subtypes.**
(JPG)

**S2 Fig. Post-dynamics analyses showing.** A) binding energy per frame and B) RMSD of the backbone atoms with respect to the initial structure of the Tat protein from different subtypes. Key: A1 (cyan) and A6 (yellow) in complex with TAR RNA over 500 ns MDS.
(JPG)

## Acknowledgments

We are thankful to Assel Berikkara, a Nazarbayev University School of Medicine graduate student who helped AZ with MD analysis.

## Author contributions

**Conceptualization:** Syed Hani Abidi.

**Funding acquisition:** Syed Hani Abidi.

**Methodology:** Alnara Zhamalbekova, Mahmoud A. A. Ibrahim, Peter A. Sidhom, Syed Hani Abidi.

**Supervision:** Syed Hani Abidi.

**Writing – original draft:** Alnara Zhamalbekova, Peter A. Sidhom.

**Writing – review & editing:** Mahmoud A. A. Ibrahim, Syed Hani Abidi.

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
