## [Decision Letter · Decision Letter 0]

20 Oct 2025

Dear Dr. Abidi,

Thank you for submitting your manuscript to PLOS ONE. After careful consideration, we feel that it has merit but does not fully meet PLOS ONE’s publication criteria as it currently stands. Therefore, we invite you to submit a revised version of the manuscript that addresses the points raised during the review process.

The writing is full of jargon that makes it difficult to go through the reading for those not expert of the topic and in particular of methods used. The study provides findings in contrast to or supporting previous ones. The authors largely discuss divergencies and similarities with previous results, losing sight of their key findings whose novelty needs to be highlighted. For this reason, I recommend trimming the discussion. The description of the results requires to be expanded and deepened. I also encourage the authors to address the concerns expressed by the reviewers, comprehensively, particularly when asked to provide additional data to support their conclusions.

We look forward to receiving your revised manuscript.

Kind regards,

Elisabetta Pilotti

Academic Editor

PLOS ONE

Journal Requirements:

This work was funded by the Nazarbayev University Faculty Development Collaborative Research Program (FDCRGP) grant No. 040225FD4710, PI Syed Hani Abidi.

Reviewers' comments:

Reviewer's Responses to Questions

**Comments to the Author**

1. Is the manuscript technically sound, and do the data support the conclusions?

Reviewer #1: Yes

Reviewer #2: No

2. Has the statistical analysis been performed appropriately and rigorously?

Reviewer #1: Yes

Reviewer #2: No

3. Have the authors made all data underlying the findings in their manuscript fully available?

Reviewer #1: Yes

Reviewer #2: Yes

4. Is the manuscript presented in an intelligible fashion and written in standard English?

Reviewer #1: Yes

Reviewer #2: Yes

Reviewer #1: The authors carried out a docking and molecular dynamic simulation of HIV Tat protein sequences from subtypes A, A1, A3, A6, B, C, D, CRF01_AE, and CRF02_A  with TAR RNA to identify subtype-specific Tat protein polymorphisms. They found that the subtypes A6, C, and CRF02_AG showed higher affinity to TAR while subtypes A3 and A1 had the weakest binding affinity to TAR with a binding energy. The increased and decreased affinity of Tat protein towards the TAR element may be attributed to subtype-specific polymorphisms suggesting that subtype-specific polymorphisms can affect Tat- TAR interactions, allowing certain subtypes to interact much more strongly with TAR as compared to others. This finding may have implications for subtype-specific disease pathogenesis mediated by the Tat protein.

However, i have few questions,

1. Have you validated subtype-specific Tat protein polymorphisms on functional cell line based assays? please refer and cite to this paper : PMID: 24465566, PMID: 28484443, PMID: 28468838 and PMID: 31652847. 2. Have you tested subtype-specific other viral genes like VIf protein polymorphisms ? please refer and cite to this paper : PMID: 264941093. does the subtype-specific Tat protein polymorphisms have any effort on host genes like CCR5? please refer and cite to this paper :PMID: 31110236

Reviewer #2: Zhamalbekova et al. present a manuscript to study the variations in HIV Tat and how 3D structures prediction, molecular docking, and molecular dynamics work. However, this study explored Tat polymorphisms and not TAR polymorphisms. Overall, the studies were performed mostly in silico, but there is a highly amount of discrepancy with previously published analyses. I believe this study should be rejected.

Major concerns: (i) A substantial body of literature exists regarding the role of the Tat-TAR axis in HIV. For instance, it demonstrates that Tat is both necessary and sufficient for HIV activity, particularly in the context of establishing and reversing latency through TAR mutations. It seems unlikely that the authors intended to overlook this prior research; discussing these earlier studies would provide important context for their results. (ii) The authors should clarify whether the TAR sequences differ among the HIV subtypes they examined. (iii) It is important to consider the disease stage of the subtypes when modeling the predictions and structures. (iv) A 200 ns molecular dynamics run may not adequately capture all relevant long-timescale conformational changes. (v) Generating consensus sequences might overlook intra-subtype diversity and rare but functionally significant variants. (vi) While the authors acknowledge their limitations in the discussion, it should be emphasized that all findings are based on in silico analyses, which can restrict direct biological or clinical relevance. They should clearly indicate that these results are predictions that require further validation.

.

Reviewer #1: No

Reviewer #2: No

---

## [Author Response · Author response to Decision Letter 1]

19 Mar 2026

Response to the Reviewers' Comments:

The authors would like to thank the Reviewers for their time and effort in reviewing our manuscript. The authors sincerely appreciate all the valuable comments and suggestions, which helped us to improve the quality of the manuscript. We have tried our best to address all comments and suggestions. The reply to each comment is given below, while the changes have been highlighted in yellow in the manuscript.

Reviewer 1

Comment #1: Have you validated subtype-specific Tat protein polymorphisms on functional cell line based assays? please refer and cite to this paper : PMID: 24465566, PMID: 28484443, PMID: 28468838 and PMID: 31652847.

Reply: We thank the reviewer for pointing out the issue about the functional validation of Tat polymorphisms. In our paper, we have incorporated findings from PMID: 28484443, PMID: 24465566, PMID: 28468838, and PMID: 31652847. Please see lines can be checked in lines 376-399.

Comment #2: Have you tested subtype-specific other viral genes like VIf protein polymorphisms ? please refer and cite to this paper : PMID: 26494109. does the subtype-specific Tat protein polymorphisms have any effort on host genes like CCR5? please refer and cite to this paper :PMID: 31110236

Reply: We thank the reviewer for suggesting these papers! As suggested, we have incorporated the findings from PMID: 31110236 and PMID: 26494109. Please see lines 92-96 and 376-380.

Reviewer 2:

Comment #1: A substantial body of literature exists regarding the role of the Tat-TAR axis in HIV. For instance, it demonstrates that Tat is both necessary and sufficient for HIV activity, particularly in the context of establishing and reversing latency through TAR mutations. It seems unlikely that the authors intended to overlook this prior research; discussing these earlier studies would provide important context for their results.

Reply: We thank the reviewer for the comment, and we agree that discussing the role of the Tat-TAR axis in latency provides important context for our findings. Thus, we have addressed this point in the revised manuscript, lines 128-136. We have reviewed key studies demonstrating Tat's role in establishing and reversing viral latency. We believe that the impact of genetic variations in Tat across subtypes (aim of our paper) beyond B and C on the Tat-TAR interaction remains underexplored.

Comment #2: The authors should clarify whether the TAR sequences differ among the HIV subtypes they examined.

Reply: We used a single and highly conserved TAR sequence across all subtypes analysis to ensure that observed differences in binding were attributable specifically to Tat protein polymorphisms. Please see lines 178-180.

Comment #3: It is important to consider the disease stage of the subtypes when modeling the predictions and structures.

Reply: We thank the reviewer for raising this point. We completely agree that disease stage is a critical factor when interpreting functional consequences of Tat polymorphisms. In the revised manuscript, we have accounted for disease stage by stratifying comparisons into acute, chronic, and AIDS stages for each subtype. We observed that different HIV-1 subtypes accumulate distinct polymorphisms as the disease progresses, as summarized in Table 2 and in the discussion section. Please see Table 2 and lines 437-442.

Comment #4: A 200 ns molecular dynamics run may not adequately capture all relevant long-timescale conformational changes.

Reply: We thank the reviewer for his valuable comment. While 200 ns molecular dynamics simulations are commonly sufficient to capture the equilibration and stability of protein-RNA complexes, the authors agree that longer simulations can further validate the robustness of the observed trends. Therefore, the authors performed extended molecular dynamics simulations of 500 ns for two representative systems: the A6-TAR complex (strongest binding) and the A1-TAR complex (weakest binding). The extended trajectories exhibited stable RMSD profiles and consistent MM-GBSA binding energies (ΔGbinding = −140.6 kcal/mol for A6-TAR and −73.4 kcal/mol for A1-TAR), in agreement with the trends observed in the 200 ns MDS. These results confirmed that the conclusions derived from the 200 ns trajectories remain valid. Please see figure S2, and lines 218-223, 290-295, and 314-318.

Comment #5: Generating consensus sequences might overlook intra-subtype diversity and rare but functionally significant variants.

Reply: We acknowledge that the consensus-sequence approach can mask specific mutations within a single subtype. To address this concern, we have built out a consensus using a threshold of more than 50% amino acid frequency at each position. We found that a substantial sequence conservation across all 9 subtypes: 90% of amino acid residues met the 50% threshold, ranging from 60-95%. The remaining amino acid residues (10%) exhibited variability, but none of these polymorphic sites corresponded to previously investigated polymorphisms that influence Tat-TAR binding affinity. Please see lines 158-165.

Comment #6: While the authors acknowledge their limitations in the discussion, it should be emphasized that all findings are based on in silico analyses, which can restrict direct biological or clinical relevance. They should clearly indicate that these results are predictions that require further validation.

Reply: We thank the reviewer for this important clarification. We agree that our study needs experimental validation. Accordingly, we have expanded the limitations section to address this point. Please see lines 453-458.

---

## [Editor Report · Decision Letter 1]

22 Mar 2026

Effect of HIV-1 subtype-specific Tat protein polymorphisms on Tat-TAR interaction

PONE-D-25-50484R1

Dear Dr. Abidi,

We’re pleased to inform you that your manuscript has been judged scientifically suitable for publication and will be formally accepted for publication once it meets all outstanding technical requirements.

Kind regards,

Elisabetta Pilotti

Academic Editor

PLOS One
---

## [Editor Report · Acceptance letter]

PONE-D-25-50484R1

PLOS One

Dear Dr. Abidi,

I'm pleased to inform you that your manuscript has been deemed suitable for publication in PLOS One. Congratulations! Your manuscript is now being handed over to our production team.

Kind regards,

on behalf of

Dr. Elisabetta Pilotti

Academic Editor

PLOS One